# SAIR: Learning Semantic-aware Implicit Representation

## Abstract

Implicit representation of an image can map arbitrary coordinates in the continuous domain to their corresponding color values, presenting a powerful capability for image reconstruction. Nevertheless, existing implicit representation approaches only focus on building continuous appearance mapping, ignoring the continuities of the semantic information across pixels. As a result, they can hardly achieve desired reconstruction results when the semantic information within input images is corrupted, for example, a large region misses. To address the issue, we propose to learn *semantic-aware implicit representation (*SAIR*)*, that is, we make the implicit representation of each pixel rely on both its appearance and semantic information (*e.g.*, which object does the pixel belong to). To this end, we propose a framework with two modules: (1) building a semantic implicit representation (SIR) for a corrupted image whose large regions miss. Given an arbitrary coordinate in the continuous domain, we can obtain its respective text-aligned embedding indicating the object the pixel belongs. (2) building an appearance implicit representation (AIR) based on the SIR. Given an arbitrary coordinate in the continuous domain, we can reconstruct its color whether or not the pixel is missed in the input. We validate the novel semantic-aware implicit representation method on the image inpainting task, and the extensive experiments demonstrate that our method surpasses state-of-the-art approaches by a significant margin.

## 1 Introduction

Recently, implicit neural representation has demonstrated surprising performance in the 2D image Chen et al. (2021b); Guo et al. (2023) and novel view Mildenhall et al. (2021); Xie et al. (2023); Zhenxing & Xu (2022) reconstruction. While existing implicit neural representation methods primarily focus on building continuous appearance mapping, they typically employ an encoder to extract appearance features from 2D images. They then utilize a neural network to associate continuous coordinates with their corresponding appearance features and translate them into the RGB color space. Unfortunately, these methods often overlook the potential semantic meaning behind the pixels, which can lead to the reconstructed result containing obvious artifacts or losing important semantic information, particularly when dealing with degraded input images, *e.g.*, a large region misses. As shown in Fig. 1, when the local appearance information is missing around the woman's eye, previous implicit representation methods like LIIF Chen et al. (2021b) fall short in accurately reconstructing the missing pixels.

To address this issue, we propose to learn semantic-aware implicit representation (SAIR), that is, we make the implicit representation of each pixel rely on both its appearance and semantic information (e.g., which object does the pixel belong to). We posit that this semantic implicit representation can significantly enhance image reconstruction quality, even when the input image is severely degraded, thereby benefiting various image processing tasks, *e.g.*, image generation, inpainting, editing, and semantic segmentation. To this end, We propose a novel approach that simultaneously leverages both continuous appearance and semantic mapping to enhance image restoration quality. This integration of continuous semantic mapping mitigates the limitations of only employing appearance implicit representation. Consequently, even in cases of degraded appearance information, the network can produce high-quality outputs with the aid of semantic information. As illustrated in Fig. 1, our method surpasses the existing implicit neural representation approaches that rely solely on appearance mapping on the image inpainting task. Remarkably, even when confronted with severely degraded

input images, *e.g.*, a large region misses, our approach still can accurately fill in the missing pixels, yielding a natural and realistic result.

The proposed semantic-aware implicit representation involved two modules: (1) building a semantic implicit representation (SIR) for a corrupted image whose large regions miss. Given an arbitrary coordinate in the continuous domain, the SIR can obtain its respective text-aligned embedding indicating the object the pixel belongs to. (2) building an appearance implicit representation (AIR) based on the SIR. Given an arbitrary coordinate in the continuous domain, AIR can reconstruct its color whether or not the pixel is missed in the input. Specifically, to implement the SIR, we first use the modified CLIP Radford et al. (2021) encoder to extract the text-aligned embedding from the input image. This specific modification (see Sec. 4.2) allows CLIP to output a spatial-aware embedding without introducing additional parameters and altering the feature space of CLIP. The text-aligned embedding can effectively reflect the pixel-level semantic information. However, this embedding has a much smaller dimension than the input image. In addition, when the input image is degraded severely, the quality of the extracted embedding is much worse. To address this problem, we utilize the semantic implicit representation within the text-align embedding. This process not only expands the feature dimensions but also compensates for missing information when the input image is severely degraded.

To implement AIR, we utilize a separate implicit representation function that takes three inputs: the appearance embedding extracted from the input image using a CNN-based network, the enhanced text-aligned embedding by SIR (see Sec. 4.3), and the pixel coordinates which indicating the location information. This allows AIR to leverage both appearance and semantic information simultaneously. As a result, even in cases of severely degraded input images, *e.g.*, large missing regions, our semantic-aware implicit representation can restore high-quality results. We validate the novel semantic-aware implicit representation (SAIR) method on the image inpainting task and conducted comprehensive experiments on the widely utilized CelebAHQ Liu et al. (2015) and ADE20K Zhou et al. (2017) datasets. The extensive experiments demonstrate that our method surpasses state-of-the-art approaches by a significant margin. In summary, our main contributions are listed as follows:

- We acknowledge the limitation of existing implicit representation methods that rely solely on building continuous appearance mapping, hindering their effectiveness in handling severely degraded images. To address this limitation, we introduce Semantic-Aware Implicit Representation (SAIR).
- We propose a novel framework to implement SAIR which involves two modules:(1) Semantic Implicit Representation (SIR) for enhancing semantic embedding, and (2) Appearance Implicit Representation (AIR), which builds upon SIR to simultaneously leverage both semantic and appearance information.
- Comprehensive experiments on the widely utilized CelebAHQ Liu et al. (2015) and ADE20K Zhou et al. (2017) datasets demonstrate that our proposed method surpasses previous implicit representation approaches by a significant margin across four commonly used image quality evaluation metrics, *i.e.*, PSNR, SSIM, L1, and LPIPS.

## 2 RELATED WORK

**Implicit neural representation.** Implicit neural functions find applications across a wide spectrum of domains, encompassing sound signals Su et al. (2022), 2D images Ho & Vasconcelos (2022); Chen et al. (2021b); Lee & Jin (2022); Ho & Vasconcelos (2022), and 3D shapes Grattarola & Vandergheynst (2022); Yin et al. (2022); Yariv et al. (2021); Hsu et al. (2021). These functions offer a means to continuously parameterize signals, enabling the handling of diverse data types, such as point clouds in IM-NET Chen & Zhang (2019) or video frames in NERV Chen et al. (2021a). Implicit neural functions have demonstrated their ability to generate novel views, as exemplified by Nerf Mildenhall et al. (2021), which leverages an implicit neural field to synthesize new perspectives. Within the domain of image processing, methods like LIIF Chen et al. (2021b) establish a connection between pixel features and RGB color, facilitating arbitrary-sized image super-resolution. LTE Lee & Jin (2022), a modification of LIIF, extends this concept by incorporating additional high-frequency information in Fourier space to address the limitations of a standalone MLP. However, these approaches lack explicit consideration of semantic information during training, which can result in potential inconsistencies at the semantic level.

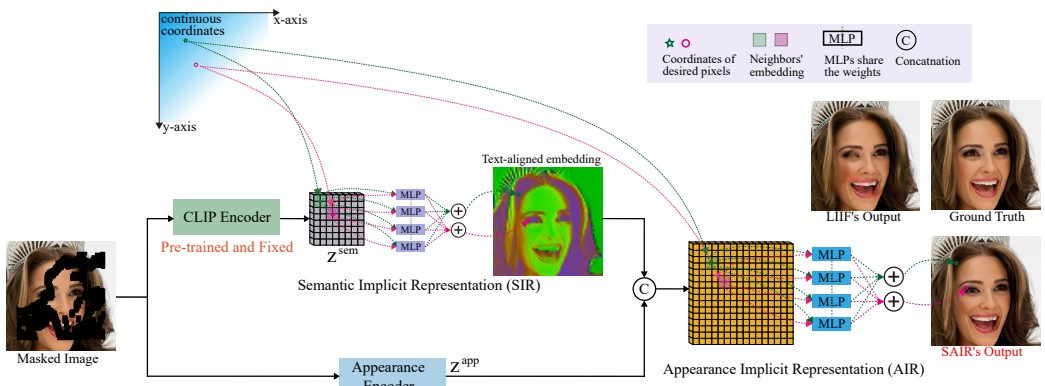

Figure 1: The overall structure of proposed semantic-aware implicit representation (SAIR). The semantic implicit representation (SIR) is used to complete the missing semantic information. The appearance implicit representation (AIR) is used to complete missing details

**Image inpainting.** Image inpainting techniques Feng et al. (2022); Bar et al. (2022); Wang et al. (2018); Li et al. (2022a); Ni et al. (2023); Zhang et al. (2023) are designed to restore corrupted image regions by leveraging information from non-missing portions. Established methods such as Ren et al. (2019); Nazeri et al. (2019); Liao et al. (2020) employ edge information or smoothed images to guide the restoration process. Another noteworthy approach, as introduced by Liu et al. (2018), relies on valid pixels to infer the missing ones. Furthermore, Guo et al. (2021) incorporates an element-wise convolution block to reconstruct missing regions around the mask boundary while utilizing a generative network to address other missing areas. Extending upon these techniques, Li et al. (2022b) advances the inpainting process by implementing element-wise filtering at both feature and image levels. Feature-level filtering is tailored for substantial missing regions, while image-level filtering refines local details. However, contemporary inpainting models face challenges when confronted with substantial missing regions, as reliable neighborhood features are often lacking. In such scenarios, text prompts prove invaluable as a robust guidance mechanism, enhancing the inpainting process.

**Image-text cross-model method.** Cross-model networks have gained substantial attention across various image processing domains, including image semantic segmentation Lüddecke & Ecker (2022); Xu et al. (2022); Zhou et al. (2023), image generation Zhu et al. (2022); Tao et al. (2022); Li et al. (2022c); Ramesh et al. (2022), and visual question answering (VQA) Zhao et al. (2022); bit (2022); Yang et al. (2021). For instance, DF-GAN Tao et al. (2022) represents a one-stage text-to-image backbone capable of directly synthesizing high-resolution images. In the realm of image segmentation, Lüddecke & Ecker (2022) leverages latent diffusion models (LDMs) to segment text-based real and AI-generated images. In VQA, Lin et al. (2022) incorporates explicit object region information into the answering model. Furthermore, Zhang et al. (2020) harnesses text to assist the model in generating missing regions within images, thereby pushing the boundaries of image inpainting tasks. Additionally, language models like CLIP Radford et al. (2021) have emerged to bridge the gap between image and semantic features. In this paper, we explore the influence of semantic information within the implicit neural function on the image inpainting task. Through the integration of semantic information, our objective is to endow the model with a more profound comprehension of the semantic meaning associated with specific image coordinates.

## 3 PRELIMINARY: LOCAL IMAGE IMPLICIT REPRESENTATION

Given an image $\mathbf{I}$, an implicit representation for the image is to map coordinates in the continuous domain to corresponding color values; that is, we have

$$c_{\mathbf{p}} = \sum_{\mathbf{q} \in \mathcal{N}_{\mathbf{p}}} \omega_{\mathbf{q}} f_{\theta}(\mathbf{z}_{\mathbf{q}}^{\text{app}}, \text{dist}(\mathbf{p}, \mathbf{q})), \tag{1}$$

where $\mathbf{p}$ is the continuous coordinates, the output $c_{\mathbf{p}}$ is the color of the pixel $\mathbf{p}$, $\mathcal{N}_{\mathbf{p}}$ contains all neighboring pixels of $\mathbf{p}$ within the image $\mathbf{I}$, $f_{\theta}(\cdot)$ is an MLP for coordinate-color mapping, $\omega_{\mathbf{q}}$ is the

weight of q, and $\mathbf{z}_{\mathbf{q}}^{\text{app}}$ is the appearance feature of pixel $\mathbf{q}$. Note that, all pixels in $\mathcal{N}_{\mathbf{p}}$ are sampled from the input image $\mathbf{I}$ and their features $\{\mathbf{z}_{\mathbf{q}}^{\text{app}}\}$ are extracted through an encoder network for handling $\mathbf{I}$. Intuitively, the MLP is to transform the appearance embedding of a neighboring pixel to the color of the pixel $\mathbf{p}$ based on their spatial distance. Recent works have demonstrated that training above implicit representation via image quality loss (*e.g.*, $L_1$ loss) could remove noise or perform super-resolution Chen & Zhang (2019); Ho & Vasconcelos (2022); Lee & Jin (2022). However, when the neighboring pixels in $\mathcal{N}_{\mathbf{p}}$ miss, the implicit representation via Eq. 1 is affected. As shown in Fig. 2, the existing implicit representation approaches cannot properly reconstruct the pixels within missing regions.

# 4 SEMANTIC-AWARE IMPLICIT REPRESENTATION (SAIR)

## 4.1 OVERVIEW

To address the issue, we propose the semantic-aware implicit representation (SAIR), which contains two key modules, *i.e.*, semantic implicit representation (SIR) and appearance implicit representation (AIR) (See Fig. 1). The first one is to build a continuous semantic representation that allows us to complete the missing semantic information within the input image. The second one is to build a continuous appearance representation that allows us to complete missing details.

Specifically, given an input image $\mathbf{I} \in \mathbb{R}^{H \times W \times 3}$ that may contain some missing regions indicated by a mask $\mathbf{M} \in \mathbb{R}^{H \times W}$, we aim to build the semantic implicit representation (SIR) to predict semantic embedding of an arbitrary given pixel whose coordinates could be non-integer values. The embedding could indicate the object the pixel belongs to. We formulate the process as

$$\mathbf{z}_{\mathbf{p}}^{\text{sem}} = \text{SIR}(\mathbf{I}, \mathbf{M}, \mathbf{p}), \tag{2}$$

where $\mathbf{z}_{\mathbf{p}}^{\text{sem}}$ denotes the semantic embedding of the pixel $\mathbf{p}$. Intuitively, we require the SIR to have three properties: ❶ The predicted semantic embedding should be well aligned with the extract category of the object the pixel belongs to. ❷ If the given coordinate (*i.e.*, $\mathbf{p}$) is within unlost regions but with non-integer values, SIR could estimate its semantic embedding accurately. This requires SIR to have the capability of interpolation. ❸ If the specified coordinate is within missing regions, SIR could complete the semantic embedding properly. We extend the local image implicit representation to the embedding level with text-aligned embeddings and propose the SIR in Sec. 4.2 to achieve the above three properties.

After getting the semantic embedding of the desired pixel, we further estimate the appearance (*e.g.*, color) of the pixel via the appearance implicit representation; that is, we have

$$c_{\mathbf{p}} = \text{AIR}(\mathbf{I}, \text{SIR}, \mathbf{p}), \tag{3}$$

where $c_{\mathbf{p}}$ denotes the color of the desired pixel $\mathbf{p}$. Intuitively, AIR is to predict the color of $\mathbf{p}$ according to the built semantic implicit representation (SIR) and input appearance. We detail the process in Sec. 4.3.

## 4.2 SEMANTIC IMPLICIT REPRESENTATION (SIR)

We first use the modified CLIP model to extract the text-aligned embedding as the semantic embedding. Specifically, inspired by the recent work MaskCLIP (Zhou et al., 2022), we remove the query and key embedding layers of the raw CLIP model and restructured the value-embedding and final linear layers into two separate $1 \times 1$ convolutional layers. This adjustment is made without introducing additional parameters or altering the feature space of CLIP, allowing the CLIP output a spatial-aware embedding tensor. Given the input image $\mathbf{I} \in \mathbb{R}^{H \times W \times 3}$, we feed it into the modified image encoder of CLIP and output a tensor $\mathbf{Z}^{\text{sem}} \in \mathbb{R}^{h \times w \times c}$ where $h$, $w$, and $c$ are the height, width, and channel numbers. Note that $\mathbf{Z}^{\text{sem}}$ is not pixel-wise embedding with $h \ll H$ and $w \ll W$, which have much lower resolution than $\mathbf{I}$. MaskCLIP employs the naive resize operation to map the $\mathbf{Z}^{\text{sem}}$ to the same size as the input image, which cannot complete the missing semantic information. Instead, we propose to extend local image implicit representation to the text-aligned embedding and formulate the SIR as

$$\mathbf{z}_{\mathbf{p}}^{\text{sem}} = \text{SIR}(\mathbf{I}, \mathbf{M}, \mathbf{p}) = \sum_{\mathbf{q} \in \mathcal{N}_{\mathbf{p}}} \omega_{\mathbf{q}} f_\theta([\mathbf{z}_{\mathbf{q}}^{\text{sem}}, \mathbf{M}[\mathbf{q}]], \text{dist}(\mathbf{p}, \mathbf{q})), \tag{4}$$

where $\mathbf{z}_{\mathbf{q}}^{\text{sem}} = \mathbf{Z}^{\text{sem}}[\mathbf{q}]$ is the embedding of the $\mathbf{q}$ location at $\mathbf{Z}^{\text{sem}}$, and $\mathcal{N}_{\mathbf{p}}$ denotes the set of neighboring coordinates around $\mathbf{p}$. $\text{dist}(\mathbf{p}, \mathbf{q})$ measures the distance between $\mathbf{p}$ and $\mathbf{q}$. $f_{\theta}(\cdot)$ is a MLP with the $\theta$ being the weights. Intuitively, $f_{\theta}(\cdot)$ is to estimate the text-aligned embedding of the location $\mathbf{p}$ according to the known embedding of $\mathbf{q}$ and the spatial relationship between $\mathbf{p}$ and $\mathbf{q}$. Finally, all estimations based on different $\mathbf{q}$ are weightly combined through $\omega_{\mathbf{q}}$ that is also set as the area ratio of $\mathbf{p}$-$\mathbf{q}$-made rectangle in the whole neighboring area.

## 4.3 Appearance Implicit Representation (AIR)

With the built SIR, we aim to build the appearance implicit representation (AIR) that can estimate the colors of arbitrarily specified coordinates. In first step, we use a CNN-based appearance encoder to generate appearance feature $\mathbf{Z}^{\text{app}} = \text{APPENCODER}(\mathbf{I}, \mathbf{M})$, and $\mathbf{Z}^{\text{app}} \in \mathbb{R}^{H \times W \times C}$. Given a pixel's coordinates $\mathbf{p}$, we predict its color by

$$c_{\mathbf{p}} = \text{AIR}(\mathbf{I}, \mathbf{M}, \text{SIR}, \mathbf{p}) = \sum_{\mathbf{q} \in \mathcal{N}_{\mathbf{p}}} \omega_{\mathbf{q}} f_{\beta}([\mathbf{z}_{\mathbf{q}}^{\text{app}}, \text{SIR}(\mathbf{I}, \mathbf{M}, \mathbf{q})], \text{dist}(\mathbf{p}, \mathbf{q})), \quad (5)$$

where $\mathbf{z}_{\mathbf{q}}^{\text{app}} = \mathbf{Z}^{\text{app}}[\mathbf{q}]$ is the appearance embedding of $\mathbf{q}$-th pixel. The function $f_{\beta}$ is a MLP with the $\beta$ being its weights. Intuitively, we estimate the color of the $\mathbf{p}$-th pixel according to the appearance and semantic information of the neighboring pixels by jointly considering the spatial distance. For example, if a pixel $\mathbf{p}$ misses, the appearance feature of $\mathbf{p}$ (*i.e.*, $\mathbf{z}_{\mathbf{p}}^{\text{app}}$) is affected and tends to zero while the semantic information could be inferred from contexts. As shown in Fig. 1, even though the pixels around the left eye miss, we still know the missed pixels belong to the left eye category.

## 4.4 Implementation Details

**Network architecture.** We utilize and modify the pre-trained ViT-B/16 image encoder of CLIP model to extract the semantic embedding. And we set the APPENCODER as a convolutional neural network and detail the architecture in Tab. 3, which is capable of generating features of the same size as the input image. Our MLP modules $f_{\alpha}(\cdot)$ and $f_{\beta}(\cdot)$ are four-layer MLP with ReLU activation layers, and the hidden dimension is 256.

**Loss functions.** During the training phase, we employ the L1 loss to measure the discrepancy between the predicted pixel color and the ground truth pixel color, which is utilized for calculating the reconstruction loss $\mathcal{L}_1$. To guarantee the feature after SIR remains in text-aligned feature space, we choose L1 loss to quantify the dissimilarity between the unmasked image's text-aligned feature $\mathbf{Z}_{\text{unmask}}^{\text{sem}} \in \mathbf{R}^{h \times w \times c}$, and the SIR reconstructed feature $\mathbf{Z}_{\text{LR}}^{\text{reconstructed}}$ without changing the resolution. The final loss function is $\mathcal{L} = \mathcal{L}_1 + \alpha \mathcal{L}_2$.

**Hyperparameters.** We employ the Adam optimizer with parameters ($\beta_1 = 0.9$, $\beta_2 = 0.999$). The learning rate is set to 0.0001 and is halved every 100 epochs. Our models are trained for 200 epochs on two NVIDIA Tesla V100 GPUs, and the batch size is set to 16.

## 5 Experimental Results

### 5.1 Setups

**Datasets.** We validate the effectiveness of proposed method through comprehensive experiments conducted on two widely used datasets: CelebAHQ Lee et al. (2020) and ADE20K Zhou et al. (2017). CelebAHQ is a large-scale dataset consisting of 30,000 high-resolution human face images, selected from the CelebA dataset Liu et al. (2015). These face images are categorized into 19 classes, and for our experiments, we use 25,000 images for training and 5,000 images for testing purposes. ADE20K, on the other hand, is a vast dataset comprising both outdoor and indoor scenes. It consists of 25,684 annotated training images, covering 150 semantic categories. We leverage this dataset to evaluate our method's performance on scene inpainting tasks. To create masked images for our experiments, we utilize the mask dataset Liu et al. (2018) similar as the previous works Li et al. (2022b). This dataset offers over 9,000 irregular binary masks with varying mask ratios, spanning from 0% to 20%, 20% to 40%, and 40% to 60%. These masks are instrumental in generating realistic inpainting scenarios for evaluation purposes.

Table 1: Comparison results on the CelebAHQ dataset across varied mask ratios.

| Method | 0%-20% | | | | 20%-40% | | | | 40%-60% | | | |
|---|---|---|---|---|---|---|---|---|---|---|---|---|
| | PSNR↑ | SSIM↑ | L1↓ | LPIPS↓ | PSNR↑ | SSIM↑ | L1↓ | LPIPS↓ | PSNR↑ | SSIM↑ | L1↓ | LPIPS↓ |
| EdgeConnect | 34.53 | 0.964 | 0.005 | 0.038 | 27.30 | 0.889 | 0.025 | 0.104 | 22.32 | 0.771 | 0.035 | 0.195 |
| RFRNet | 34.93 | 0.966 | 0.005 | 0.035 | 27.50 | 0.890 | 0.024 | 0.100 | 22.77 | 0.775 | 0.033 | 0.185 |
| JPGNet | 35.86 | 0.972 | **0.004** | 0.040 | 28.18 | 0.909 | 0.023 | 0.119 | 22.32 | 0.771 | 0.035 | 0.195 |
| LAMA | 36.04 | 0.973 | 0.008 | 0.024 | 29.14 | 0.932 | 0.020 | 0.029 | 22.94 | 0.854 | 0.033 | 0.152 |
| MISF | 36.32 | 0.976 | 0.012 | 0.019 | 29.85 | 0.932 | 0.021 | 0.055 | 23.91 | 0.868 | 0.042 | 0.133 |
| LIIF | 35.27 | 0.969 | 0.012 | 0.023 | 28.80 | 0.923 | 0.026 | 0.043 | 23.30 | 0.830 | 0.051 | 0.136 |
| SAIR | **37.97** | **0.977** | 0.010 | **0.016** | **31.49** | **0.944** | **0.019** | **0.025** | **24.87** | **0.870** | **0.031** | **0.124** |

Table 2: Comparison results on the ADE20K dataset across varied mask ratios.

| Method | 0%-20% | | | | 20%-40% | | | | 40%-60% | | | |
|---|---|---|---|---|---|---|---|---|---|---|---|---|
| | PSNR↑ | SSIM↑ | L1↓ | LPIPS↓ | PSNR↑ | SSIM↑ | L1↓ | LPIPS↓ | PSNR↑ | SSIM↑ | L1↓ | LPIPS↓ |
| EdgeConnect | 30.91 | 0.948 | 0.007 | 0.049 | 24.18 | 0.841 | 0.022 | 0.139 | 20.07 | 0.694 | 0.048 | 0.259 |
| RFRNet | 30.36 | 0.937 | 0.008 | 0.073 | 23.42 | 0.807 | 0.027 | 0.199 | 19.21 | 0.638 | 0.060 | 0.340 |
| JPGNet | 31.65 | 0.952 | 0.007 | 0.074 | 24.72 | 0.851 | 0.022 | 0.202 | 20.46 | 0.713 | 0.048 | 0.342 |
| LAMA | 31.07 | 0.956 | 0.009 | 0.036 | 24.15 | 0.859 | 0.025 | 0.116 | 20.15 | 0.713 | 0.048 | 0.257 |
| MISF | 31.45 | 0.954 | 0.006 | **0.032** | 24.97 | 0.859 | **0.020** | 0.117 | 20.59 | 0.717 | 0.044 | 0.233 |
| LIIF | 30.96 | 0.946 | 0.010 | 0.038 | 24.57 | 0.846 | 0.026 | 0.120 | 19.79 | 0.708 | 0.049 | 0.274 |
| SAIR | 31.01 | **0.964** | **0.005** | 0.034 | **26.44** | **0.866** | 0.023 | **0.110** | **21.88** | **0.722** | **0.042** | **0.193** |

**Baselines.** We enhance our approach by incorporating semantic representations based on previous implicit neural function model LIIF Chen et al. (2021b). By modifying image encoder and integrating semantic information, we obtain the semantic-aware implicit function, denoted as SAIR. For comparative analysis, we select state-of-the-art inpainting methods StructFlow Ren et al. (2019), EdgeConnect Nazeri et al. (2019), RFRNet Li et al. (2020), JPGNet Guo et al. (2021), LAMA Suvorov et al. (2022), MISF Li et al. (2022b), and the implicit neural function without semantic information LIIF Chen et al. (2021b) as our baselines.

**Evaluation metrics.** To assess the performance of all methods, we utilize four commonly employed image quality evaluation metrics: peak signal-to-noise ratio (PSNR), structural similarity index (SSIM), L1 loss, and learned perceptual image patch similarity (LPIPS) Zhang et al. (2018). PSNR, SSIM, and L1 offer insights into the quality of the generated image, while LPIPS quantifies the perceptual distance between the restored image and the ground truth.

## 5.2 COMPARISON RESULTS

The results obtained on the CelebAHQ dataset are presented in Tab. 1, demonstrating a significant performance improvement achieved by incorporating semantic information into the models. For instance, SAIR outperforms MISF by 1.74 in PSNR for the 0–20% mask ratio. Moreover, SAIR surpasses LAMA by 2.35 in PSNR and 1.2% in SSIM for 20–40% ratio. In the 20–40% mask ratio, SAIR exhibits enhancements of 2.69 in PSNR and 7.1% in SSIM compared to LIIF. The results on the ADE20K dataset, as shown in Tab. 2, also reveal the effectiveness of incorporating semantic information. SAIR achieves a lowered LPIPS of 0.193 for the 40–60% mask ratio. And SAIR improves PSNR to 26.44 and SSIM to 86.6% in 20–40% ratio range. Notably, SAIR attains the best PSNR and SSIM performance for all mask ratios. These results demonstrate that semantic information aids in processing degraded images. Our approach overcomes the limitations imposed by noise in masked area appearance features by leveraging the guidance of semantic information.

Qualitative results from different models are presented in Fig. 2, showcasing significant enhancements achieved by our proposed method. Fig. 2 unmistakably illustrates that the implicit neural function models lacking semantic guidance tend to produce blurry reconstructions in the affected regions, often displaying a noticeable boundary between masked and unmasked areas. In contrast, models enriched with semantic information yield more visually coherent and pleasing results. As observed in the first row, it becomes apparent that traditional implicit neural functions like LIIF struggle to recover

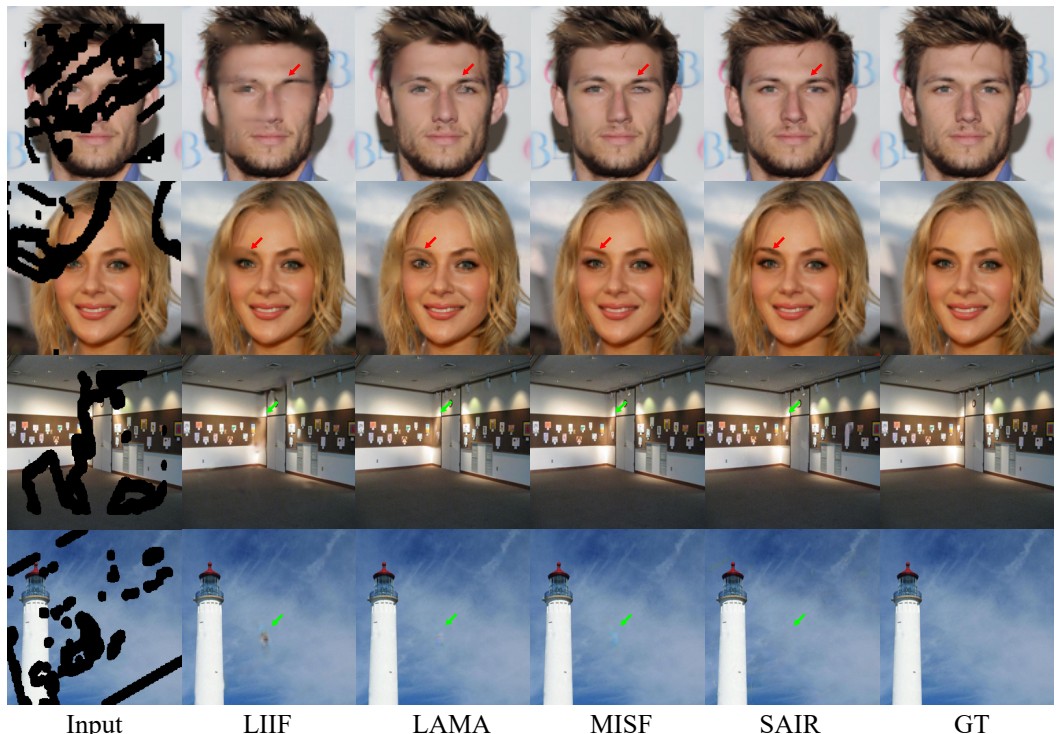

| Input | LIIF | LAMA | MISF | SAIR | GT |

Figure 2: Visual comparison with competitors: the first two cases are from the CelebAHQ dataset, while the last two are from the ADE20K dataset.

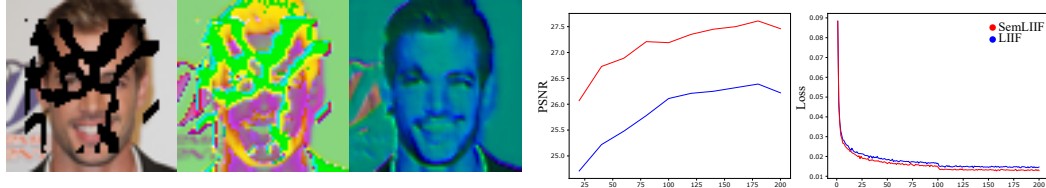

Figure 3: From left to right, we show the input masked image, masked semantic feature after CLIP encoder, and semantic feature after SIR.

Figure 4: PSNR vs Epoch and Training Loss vs Epoch.

the 'eye' category when it is entirely masked. In such cases, the neighboring appearance features can only provide information about the 'face.' However, SAIR demonstrates its ability to reconstruct the 'eye' category effectively, benefitting from the restored semantic features. Furthermore, in the last row of Fig. 2, the original implicit neural function generates unexpected regions prominently.

In Fig. 3, we present a visual representation of the image features before and after the application of our SIR module. The pre-trianed CLIP encoder cannot handle the masked regions ideally. And it becomes evident that our proposed SIR module effectively reconstructs the corrupted image feature. To assess the impact of semantic information during the training process, we visually analyze the training progress of both LIIF and SAIR. The training loss curves depicted in Fig. 4 demonstrate that both models converge at a similar point. This observation suggests that the inclusion of semantic information can facilitate loss convergence without necessitating an extended training duration. Moreover, as seen in Fig. 4, the PSNR curve illustrates that the model enriched with semantic information consistently outperforms the original implicit representation model right from the outset.

### 5.3 ABLATION STUDY AND DISCUSSION

**Study on using different image encoders.** To demonstrate the compatibility of our semantic feature embedding with various image encoders, we conducted an ablation study in which we replaced our image encoder with the original LIIF encoder EDSR Lim et al. (2017). As indicated in Tab. 4,

Table 3: Architecture of APPENCODER. H × W is the resolution of the input image.

| Output size | Operation |
|---|---|
| H × W | Conv(4, 64, 7, 1, 3), ReLU |
| H/2 × W/2 | Conv(64, 128, 4, 2, 1), ReLU |
| H/4 × W/4 | Conv(128, 256, 4, 2, 1), ReLU |
| Resnet × 8 | |
| H/4 × W/4 | Conv(256, 256, 3, 1, 1), ReLU |
| H/4 × W/4 | Conv(256, 256, 3, 1, 1), ReLU |
| H/2 × W/2 | Conv(256, 128, 4, 2, 1), ReLU |
| H × W | Conv(128, 64, 4, 2, 1), ReLU |

Table 4: Ablation study results on different image encoders and different implicit neural function models on CelebAHQ dataset.

| Variant | All mask ratios | |
|---|---|---|
| | PSNR↑ | SSIM↑ |
| EDSR(wo) | 30.26 | 0.892 |
| EDSR(w) | **31.48** | **0.913** |
| LTE | 30.60 | 0.931 |
| SemLTE | **31.97** | **0.939** |

when compared to a model without the inclusion of semantic features (EDSR(wo)), the model that incorporates semantic features (EDSR(w)) also exhibited improvements, increasing the PSNR by 1.12 and the SSIM by 2.1%. These experiments provide compelling evidence that semantic information has the potential to enhance performance across different appearance feature spaces.

**Study on using different implicit neural functions.** In order to demonstrate the versatility of our semantic feature integration with various implicit neural functions, we conducted an ablation study using another implicit neural function known as LTE Lee & Jin (2022), which is specifically designed for image super-resolution tasks. In this study, we seamlessly incorporated semantic features into LTE, creating what we refer to as SemLTE. The resulting performance metrics are presented in Tab. 4, where SemLTE achieved significant improvements, elevating the PSNR to 31.97 and the SSIM to 93.9%. These outcomes affirm the adaptability of our proposed semantic implicit representation, showcasing its effectiveness when applied to different implicit neural functions.

**Study on the models with/without SIR block.** To further assess the effectiveness of our proposed SIR module, we conducted performance tests on the CLIP encoder, both with and without the SIR in the semantic segmentation task. We use masked images as inputs to generate the segmentation results, which are compared with ground truth. In the setting 'without SIR', we initially employed the CLIP text encoder to produce category features $\mathbf{CLIP\_T} \in \mathbb{R}^{L \times C}$ for all categories in the dataset, where L represents the number of categories. Subsequently, we used $\mathbf{CLIP\_T}$ to filter the semantic feature $\mathbf{CLIP\_I}$, yielding a pixel-wise segmentation map $\mathbf{S} \in \mathbb{R}^{H \times W \times L}$. In the setting 'with SIR', we use the SIR block to reconstruct the CLIP semantic feature $\mathbf{CLIP\_I}$, the reconstructed feature is used for segmentation. The results presented in Tab. 5 indicate that the inclusion of the SIR block leads to a notable increase in mIoU by 0.28, demonstrating the effective capacity of the SIR model to reconstruct semantic features.

**Study on not filling the semantic feature (NFS).** In the preceding section, we employed SIR to reconstruct the semantic feature of masked images. Here, we delve into an alternative scenario where we do not to fill in the masked semantic feature. In our experiments, we introduced masked semantic features into the implicit neural function alongside the appearance feature. However, as evident in the results presented in Tab. 6 under the label NFS, this approach yields suboptimal performance when compared to SAIR. Specifically, it leads to a noticeable decrease of 2.04 in PSNR and a 2.1% reduction in SSIM. The presence of meaningless semantic information within the masked region exerts an adverse influence on the construction of the implicit representation.

**Study on only using semantic feature to build implicit representation (OUS).** In this section, we explore the possibility of constructing a continuous representation using only semantic features, meaning that we exclusively input semantic information into the implicit neural function. The results is shown in Tab. 6 as the OUS. It's worth noting that the CLIP image encoder is trained to produce features that align with textual information. In essence, this experiment underscores the significance of integrating both semantic and image-level information to attain favorable outcomes in image generation tasks.

**Using other semantic embeddings.** As an alternative to employing our Semantic Implicit Representation (SIR), we can also utilize existing models designed for semantic embeddings, such as the previously introduced semantic segmentation model SAM Kirillov et al. (2023). To demonstrate this, we replaced our CLIP encoder with the pre-trained SAM image encoder, and the results are presented in Tab. 6. Notably, it becomes evident that the CLIP encoder outperforms traditional

Table 5: Semantic segmentation results from the models with/without SIR on ADE20K dataset.

| Variant | mIoU |
|---|---|
| CLIP Encoder | 0.17 |
| CLIP Encoder+ SIR | **0.45** |

Table 6: Ablation study results on *Not filling semantic feature* (NFS), *Only using semantic feature* (OUS), and SAM encoder on CelebAHQ dataset.

| Variant | All mask ratios | |
|---|---|---|
| | PSNR↑ | SSIM↑ |
| NFS | 30.32 | 0.923 |
| OUS | 31.11 | 0.929 |
| SAM Encoder | 31.72 | 0.935 |
| SAIR | **32.36** | **0.944** |

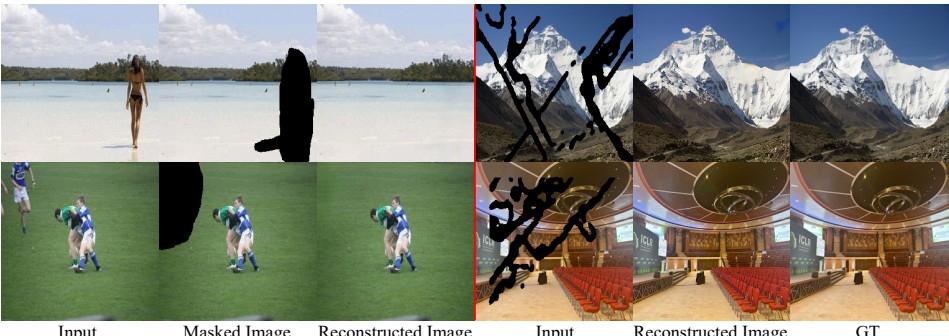

| Input | Masked Image | Reconstructed Image | Input | Reconstructed Image | GT |

Figure 5: Real-life applications using in-the-wide images. We show object removal results (left) and inpainting results (right).

semantic segmentation encoders in this context. This superiority is attributed to the CLIP encoder's capacity to capture rich textual information, further enhancing the inpainting task's performance.

**Real-life applications.** Our approach demonstrates versatile applications, including object removal and image restoration. To test the performance of our model in real-life applications, we use our model trained on ADE20K dataset to process in-the-wild images selected from the Internet. As shown in Fig. 5, our method consistently delivers promising results in addressing the challenges posed by diverse and uncontrolled in-the-wild scenarios.

## 6 CONCLUSION

In this paper, we tackle the limitations inherent in existing implicit representation techniques, which predominantly rely on appearance information and often falter when faced with severely degraded images. To address this challenge, we introduce a novel approach: the learning of a semantic-aware implicit representation (SAIR). By seamlessly using a semantic implicit representation (SIR) to handle the pixel-level semantic feature and a appearance implicit representation (AIR) tp reconstruct the image colour, our method effectively mitigates the impact of potentially degraded regions. To gauge the effectiveness of our approach, we conducted comprehensive experiments on two widely recognized datasets, CelebAHQ Liu et al. (2015) and ADE20K Zhou et al. (2017). The results unequivocally demonstrate that our method outperforms existing implicit representation and inpainting approaches by a substantial margin across four commonly employed image quality evaluation metrics. Our model's capacity to assist the implicit neural function in processing damaged images expands its utility and applicability, offering promising prospects for various image-related tasks.

**Limitations.** In this study, we have showcased the effectiveness of the semantic-aware implicit representation within the domain of image inpainting. While our proposed method has demonstrated remarkable performance in this particular task, its broader applicability across other vision-related tasks has yet to be fully explored. As part of our future research endeavors, we plan to conduct additional experiments to assess the potential of our method in addressing various vision tasks beyond inpainting.

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
