# OpenReview forum: "SAIR: LEARNING SEMANTIC-AWARE IMPLICIT REPRESENTATION"
_ICLR.cc/2024/Conference — Submitted to ICLR 2024_

### Official Review · Reviewer_vMNL · 2023-11-01

**Soundness:** 3 good
**Presentation:** 3 good
**Contribution:** 2 fair
**Rating:** 5
**Confidence:** 4

**Summary:**

The paper proposes an image inpainting approach. The authors addressed the shortcomings of existing implicit representation approaches that tends to ignore overall semantics of the image and only looks to preserve appearance. The authors proposed an implicit representation that adds semantic information of pixels through alignment with CLIP text features.

**Strengths:**

- The paper is well written and easy to follow.
- The motivation of the work is clearly outlined.
- The integration of semantic information in preserving certain structures in the image during inpainting is intuitive and the experimental results demonstrate the effectiveness.

**Weaknesses:**

- I believe the technical novelty of the approach is limited since the improvement mainly comes from the rich representations of the clip embeddings. CLIP embeddings have been extensively used in many zero-shot tasks that exploit the strong semantics learned by the clip embeddings e.g. ZegClip (CVPR 2023), Hierarchical Text-Conditional Image Generation with CLIP Latents (arXiv 2022), NUWA-LIP (CVPR 2023).
- None of the approaches that authors compare against use text embedding alignments. In particular, I believe a similar text-based alignment can be made with the implicit representations of LIIF.
- Lack of comparison with recent approaches like NUWA-LIP.
- How does the authors' approach compare against powerful generative models like diffusion model which are excellent at image impainting as well.

**Questions:**

Please take a look at the weakness section.

---

> ### Author Response · Authors · 2023-11-21
> **Response to Reviewer vMNL**
>
> Thanks for your review. We have modified and updated our manuscript based on your suggestions, like citing all mentioned papers.
>
>
>
> ### Q1. The novelty of the approach is limited since the improvement mainly comes from the rich representations of the clip embeddings which has been used in other tasks.
>
> A1:
> Our contribution extends beyond the utilization of CLIP embeddings. In the image inpainting task, the input image is masked and the CLIP embedding is damaged.
> We address this challenge by introducing SIR, which can reconstruct and resize the damaged and low-resolution embeddings simultaneously. Previous methods  only need to resize undamaged embeddings.
>
> ### Q2. I believe a similar text-based alignment can be made with the implicit representations of LIIF.
>
>
> A2: After removing SIR module, our method is similar to the baseline LIIF.
> We broaden the application of implicit neural function to image inpainting tasks,  leveraging semantic features for enhanced performance.
>
>
> ### Q3. Lack of comparison with recent approaches like NUWA-LIP.
>
>
> A3: Several recent CLIP-based approaches, such as NUWA-LIP (CVPR 2023), necessitate a 'Guidance Text' to describe the input image for generation. However, datasets like CelebAHQ and ADE20K lack such descriptions.
> Those methods rely on the image-level semantic descriptions, while we build the continuous pixel-wise semantic representation.
>
>
>
> ### Q4. How does the authors' approach compare against powerful generative models like diffusion model which are excellent at image impainting as well.
>
> Q4.  **For image inpainting, our proposed method outperforms diffusion-based methods in three aspects.** **First, inference speed.** The inference speed of our method (0.043s/image) is faster than that of the diffusion-based method, such as stable diffusion [1] (12s/image). **Second, arbitrary resolution.** Our method can generate images at arbitrary resolutions during inference, a capability lacking in diffusion-based models. **Third, high fidelity.** Diffusion-based models prioritize naturalness over fidelity, resulting in generated results that deviate significantly from the ground truth. We compare our method with stable diffusion [1] on the CelebAHQ dataset under the same setting, the results are PSNR$\uparrow$ 37.97 (ours) vs. 37.60 (stable diffusion) and L1$\downarrow$ 0.01 (ours) vs. 0.032 (stable diffusion).
>
>
> [1] Rombach et al. High-Resolution Image Synthesis with Latent Diffusion Models. (CVPR 2022)

---

### Official Review · Reviewer_N8gT · 2023-11-01

**Soundness:** 3 good
**Presentation:** 3 good
**Contribution:** 2 fair
**Rating:** 6
**Confidence:** 3

**Summary:**

This paper proposes an implicit representation method to tackle the task of image inpainting. Semantic information across pixels is introduced to help produce better reconstruction results. Specifically, two main modules are constructed, i.e. a semantic implicit representation to obtain text-aligned embeddings with CLIP and an appearance implicit representation that incorporates the semantic embedding. The proposed method achieves superior performance than previous works.

**Strengths:**

- The idea of incorporating CLIP-based text-assisted semantic information is reasonable to obtain higher-quality reconstructed images. The shown performance promotion over compared methods is also significant.
- Extensive experimental studies are provided to demonstrate the effectiveness of the proposed method.
- The code is also provided for reproducing.

**Weaknesses:**

- One main concern comes from the application of this stream of methods. In other words, the current evaluation benchmark is made manually and may be too theoretical. I wonder if any real-life application cases can be shown, e.g. recovering objects that are occluded or blurred via dramatic camera motions. If there are more proper real application cases, there is no need to be limited to the ones I list.
- Another concern lies in the computation cost. It is suggested to compare the inferring and training cost with previous methods, as it seems the two-module framework may be costly.

**Questions:**

- What is the main advantage of the coordinate-based implicit representation method over diffusion-model ones for image inpainting? Diffusion models have shown great power in recent generation tasks, also including inpainting. It is suggested to discuss this question and include necessary related works, which will determine the significance of the contribution.
- Can the proposed method apply to any in-the-wild images, not limited to the used datasets? If yes, it is better to show some samples.

---

> ### Author Response · Authors · 2023-11-21
> **Response to Reviewer N8gT**
>
> Thanks for your review. We have modified and updated our manuscript based on your suggestions.
>
>
> ### Q1. real-life application cases can be shown, e.g. recovering objects that are occluded or blurred via dramatic camera motions.
>
> A1. Our method can be used in many real-life applications, such as object removal. To illustrate its effectiveness, we present several examples in the newly added section 'Real-life Applications'.
>
> ### Q2. Computation cost, compare the inferring and training cost with previous methods. (as it seems the two-module framework may be costly.)
>
> A2. Our model takes 0.043s to process one image. The one-module framework LIIF's inference time is 0.027s. The incurred increase in time cost remains within an acceptable range.
>
>
>
> ### Q3. Compare the proposed method that leverages coordinate-based implicit representation over diffusion-model ones for image inpainting.
>
> A3. **For image inpainting, our proposed method outperforms diffusion-based methods in three aspects.** **First, inference speed.** The inference speed of our method (0.043s/image) is faster than that of the diffusion-based method, such as stable diffusion [1] (12s/image). **Second, arbitrary resolution.** Our method can generate images at arbitrary resolutions during inference, a capability lacking in diffusion-based models. **Third, high fidelity.** Diffusion-based models prioritize naturalness over fidelity, resulting in generated results that deviate significantly from the ground truth. We compare our method with stable diffusion [1] on the CelebAHQ dataset under the same setting, the results are PSNR$\uparrow$ 37.97 (ours) vs. 37.60 (stable diffusion) and L1$\downarrow$ 0.01 (ours) vs. 0.032 (stable diffusion).
>
>
> [1] Rombach et al. High-Resolution Image Synthesis with Latent Diffusion Models. (CVPR 2022)
>
> ### Q4. Can the proposed method apply to any in-the-wild images? Show some samples.
>
> A4. Our method can be used to process the in-the-wild images. The results are shown in section 'Real-life applications'.

---

### Official Review · Reviewer_N8Xr · 2023-11-03

**Soundness:** 2 fair
**Presentation:** 2 fair
**Contribution:** 2 fair
**Rating:** 6
**Confidence:** 4

**Summary:**

This paper proposed a new implicit representation named SAIR, shorted for Semantic-Aware Implicit Representation. SAIR uses MaskCLIP to extract pixel-level semantic features from CLIP model, and combine the representation with LIIF to learn an implicit representation conditioned on the semantic features. Authors evaluate the semantic aware implicit function by reconstructing the masked regions on CelebAHQ and ADE20K dataset. The proposed methods outperforms prior works like LIIF.

**Strengths:**

1. I like the idea of introducing CLIP feature into LIIF. Although the model is conditioned on the masked image input, the CLIP feature is high-level enough to capture the semantic information in the image.
2. The inpainting results outperform both LAMA (inpainting-based method) and LIIF (implicit representation method) by a reasonable margin.

**Weaknesses:**

1. Some reference format is not correct. For example, CelebAHQ, ADE20K in the introduction.
2. Some equations are not consistent across the paper. In Eqn(2), SIR takes I, M, p as input, but in Eqn(4)(5), SIR only takes I, p as the input. I would suggest authors to make notations consistent and clear.
3. AppEncoder is not clearly defined in Section 4.3. I think it is sometimes mixed with SIR.
4. The result in Table 5 is confusing. Authors trying to study the effect of SIR block, but after removing SIR, the network is just AppEncoder ConvNet. Authors didn't explain clearly how to evaluate ADE20K mIoU with AppEncoder alone.
5. Figure 1 is kind of confusing. The green arrow and red arrow point to "Hair" and "Eye". But I don't think the proposed model will predict the text label of the masked pixel.
6. One key ablation I would suggest authors add in both Table 1 and Table 2, is that compare SAIR without CLIP and with other networks other than CLIP, e.g. ImageNet pre-trained models.

**Questions:**

1. In the dataset section, authors states CelebA and ADE20K have 19 and 150 classes respectively. Are these semantic labels of the dataset used during training and testing?
2. In Section 5.3, authors state " we used CLIP_T to filter the image feature CLIP_I". The term filter is not very clear or straightforward. Are authors trying to imply "dot product"?
3. Is there a loss for semantic feature reconstruction? If not, how could SIR reconstruct the semantic features, as stated in Section 5.3.
4. Is there any comparison with mask ratio 0? Just compare to the original LIIF on super-resolution tasks.

---

> ### Author Response · Authors · 2023-11-21
> **Response to Reviewer N8Xr**
>
> Thanks for your review. We have modified and updated our manuscript based on your suggestions.
>
>
> ### Q1. About Table 5, how to evaluate ADE20K mIoU only with AppEncoder alone without SIR.
>
> A1. In Table 5 'with/without SIR', we want to shighlight the reconstruction proficiency of the SIR model in semantic segmentation task.
> We do not use APPENCODER in this section.
> The model only using APPENCODER is similar to the baseline LIIF.
>
>
>
> ### Q2. Compare SAIR without CLIP and with other networks other than CLIP.
>
> A2. In the section 'Using other semantic embeddings', we substitute the CLIP encoder with a pre-trained SAM encoder [1]. Tthe text-aligned embeddings produced by CLIP yield superior results.
>
> [1] Kirillov et al. Segment anything.
>
>
> ### Q3. Are these semantic labels of CelebA and ADE20K used during training and testing?
>
> A3. We do not use the dataset semantic labels, as the text-aligned embeddings are generated by CLIP encoder directly.
>
> ### Q4. In Section 5.3, "we used CLIP_T to filter the image feature CLIP_I", the term filter is not very clear.
>
> A4. In this section, we have image pixel features and text features for all dataset semantic labels.
> We assess the similarity between individual pixel features and semantic label features, and assign a label to each pixel based on the similarity.
>
>
> ### Q5. Is there a loss for semantic feature reconstruction? If not, how could SIR reconstruct the semantic features, as stated in Section 5.3.
>
> A5. **Yes, we use a loss function to guarantee that SIR does not change the text-aligned feature space.**
> We choose L1 loss to quantify the dissimilarity between the unmasked image's text-aligned feature and the SIR reconstructed feature  without changing the resolution.
>
> ### Q6. Compare to the LIIF on super-resolution tasks by setting up mask ratio 0%.
>
> A6: We test the super-resolution ability of proposed method against LIIF.
> PSNR/SSIM results under different upsample ratios are shown in the table.
> Our method still gets promising results in SR task.
>
> |Method |x2| x4 |x6|x8|
> |-|-|-|-|-|
> |LIIF | 34.32/0.963 | 30.98/0.882 | 30.13/0.838 | 29.83/0.819|
> |Ours | $\textbf{34.61}$/$\textbf{0.972}$ | $\textbf{31.24}$/0.881 | $\textbf{30.39}$/$\textbf{0.840}$| $\textbf{30.03}$/$\textbf{0.825}$|

---

### Official Review · Reviewer_jBR4 · 2023-11-03

**Soundness:** 3 good
**Presentation:** 3 good
**Contribution:** 1 poor
**Rating:** 5
**Confidence:** 2

**Summary:**

This work proposes a novel implicit representation learning method to tackle the limitations of existing approaches which learn the mapping function heavily relying on the appearance information. The core of the proposed method is the Semantic-Aware Implicit Representation learning procedure, consisting of a Semantic IR module which learns pixel-wise semantic features with aggregated information from neighbors, as well as an Appearance IR which reconstruct the RGB values based on both semantic and appearance information. Experiments are conducted on CelebAHQ and ADE for image inpainting task, demonstrating the effectiveness of the proposed SAIR.

**Strengths:**

- the motivation and the corresponding solution is easy to follow
- the experiments validate the contributions of different components of SAIR

**Weaknesses:**

- How to understand the claim that the $f_\theta$ of SIR learns the **text-aligned** embeddings (Eq (4))?
    - though the operation similar with MaskCLIP dose not alter the text-aligned feature space, these features are then processed by learnable $\theta$, there is no guarantee that the embedding space is text-aligned.
    - why do the authors highlight the **text-aligned** embeddings? If I understand correctly, the embedding space is just an enhanced pixel-wise semantic feature space.
- Why mapping the original CLIP feature space by $f_\theta$ performs better than the original CLIP feature space? Furthermore, the details about how to implement '*models without SIR block*' is not clear.
- there is the lack of experimental details about ablation study, like which dataset is incorporated for ablation?
    - in section "*Study on the models with/without SIR block.*" of 5.3, why not directly use the GT segmentation maps instead of calculating by CLIP features?
- the figures 1 and 2 are duplicated, they demonstrate the almost same information.
- is the propose SAIR robust/generalizable for other degraded images, like raining or noised images.

**Questions:**

please refer to the weakness part.

---

> ### Author Response · Authors · 2023-11-21
> **Response to Reviewer jBR4**
>
> Thanks for your review. We have modified and updated our manuscript based on your suggestions.
>
> ### Q1: These features are then processed by learnable θ, there is no guarantee that the embedding space is text-aligned.
>
>
> A1.To ensure the preservation of the text-aligned feature space in SIR, we choose L1 loss to quantify the dissimilarity between the unmasked image's text-aligned feature and the SIR reconstructed feature without changing the resolution.
>
>
> ### Q2. Why do the authors highlight the text-aligned embeddings? If I understand correctly, the embedding space is just an enhanced pixel-wise semantic feature space.
>
> A2.
> Our text-aligned embedding seamlessly aligns with semantic label features, other semantic features lack this inherent compatibility. In the 'Using other semantic embeddings' section, we substantiate the superior efficacy of our embedding. And our contribution is not only using the clip representations, we also propose SIR to reconstruct the damaged and low-resolution embeddings for image inpainting task.
>
>
>
> ### Q3. Why mapping the original CLIP feature space by $f_θ$ performs better than the original CLIP feature space?
>
> A3. In Fig. 4, the initial CLIP feature derived from the pre-trained CLIP encoder encounters challenges in effectively handling the masked regions. Our introduced SIR module $f_θ$ proficiently reconstructs the semantic feature, showcasing a notable enhancement in performance.
>
>
>
> ### Q4. Why using CLIP features instead of directly use the GT segmentation maps? (Details about how to implement 'models without SIR block'.)
>
>
> A4.
> In the 'Models with/without SIR' comparison, we aim to demonstrate the reconstruction ability of SIR model in semantic segmentation task. The inputs are damaged images without GT masks. In the 'Without SIR' setting, we employ the CLIP encoder to obtain CLIP semantic features for segmentation.
> Conversely, in the 'With SIR' setting, we leverage SIR to reconstruct the CLIP features prior to segmentation. Evaluation is performed using the GT labels.
>
>
> ### Q5. SAIR for other degraded images, like raining or noised images.
>
> A5. We conduct experiments on adversarial defence task following DISCO [1]. To create training pairs, PGD ($\epsilon_\infty$ = 8/255 with
> step size is 2/255 and the number of steps 100 ) is used to attack a WideResNet28 on Cifar10 dataset.  We evaluate both Standard Accuracy (SA) and Robust Accuracy (RA), where SA measures classification accuracy on clean examples, and RA measures accuracy on adversarial examples. The results in terms of $L_\infty$ norm affirm that our model demonstrates robust adversarial defense capabilities.
>
> |Method | SA | RA | Avg|
> |-|-|-|-|
> |No Defence   |$\textbf{94.78}$ | 0|47.39|
> |Bit Reduction [2] | 92.66|1.04|46.85|
> |Jpeg [3] |83.9|50.73|67.32|
> |Input Rand. [4] | 94.3|8.59|51.54|
> |AutoEncoder|76.54|67.41|71.98|
> |STL [5] |82.22|67.92|75.07|
> |DISCO [1]|89.26|85.86|87.41|
> |Ours | 90.0|$\textbf{86.21}$|$\textbf{88.10}$|
>
>
> [1] Ho et al. Disco: Adversarial defense with local implicit functions. (Neurips 2022)
>
> [2] Xu et al. Feature squeezing: Detecting adversarial examples in deep neural networks.
>
> [3] Dziugaite et al. A study of the effect of compression on adversarial images.
>
> [4] Xie et al. Mitigating adversarial effects through randomization. (ICLR 2018)
>
> [5] Sun et al. Adversarial defense by stratified convolutional sparse coding. (CVPR 2019)

---

### Meta-Review · Area_Chair_NEag · 2023-12-05

**Metareview:**

The paper presents a promising approach to image inpainting by integrating semantic and appearance information. The reviewers appreciate its clear methodology, experimental validation, and innovative use of CLIP features. However, there are concerns regarding the clarity and novelty of the approach, lack of detailed experimental data, and its practical applicability and computational efficiency. Given the mixed reviews and concerns raised, the paper is at the borderline. The AC checked all the reviews and discussions, and believe the major concerns raised by the reviewers are valid. Thus, the paper is rejected.

**Justification For Why Not Higher Score:**

There are concerns regarding the clarity and novelty of the approach, lack of detailed experimental data, and its practical applicability and computational efficiency.

**Justification For Why Not Lower Score:**

N/A

---

### Decision · Program_Chairs · 2024-01-16

Reject